# Blood Serum Stimulates the Virulence Potential of Mucorales through Enhancement in Mitochondrial Oxidative Metabolism and Rhizoferrin Production

**DOI:** 10.3390/jof9121127

**Published:** 2023-11-22

**Authors:** José Alberto Patiño-Medina, Viridiana Alejandre-Castañeda, Marco Iván Valle-Maldonado, Mauro Manuel Martínez-Pacheco, León Francisco Ruiz-Herrera, Joel Ramírez-Emiliano, Oscar Abelardo Ramírez-Marroquín, Karla Viridiana Castro-Cerritos, Jesús Campos-García, Martha Isela Ramírez-Díaz, Victoriano Garre, Ulrike Binder, Víctor Meza-Carmen

**Affiliations:** 1Instituto de Investigaciones Químico Biológicas, Universidad Michoacana de San Nicolás de Hidalgo, Morelia 58030, Mexico; jpatino@umich.mx (J.A.P.-M.); viridiana.alejandre@umich.mx (V.A.-C.); mauro.martinez.pacheco@umich.mx (M.M.M.-P.); leon.ruiz@umich.mx (L.F.R.-H.); jesus.campos@umich.mx (J.C.-G.); martha.ramirez@umich.mx (M.I.R.-D.); 2Laboratorio Estatal de Salud Pública del Estado de Michoacán, Morelia 58279, Mexico; mimaldonado@umich.mx; 3Departamento de Ciencias Médicas, Universidad de Guanajuato, León 37320, Mexico; joelre@ugto.mx; 4Instituto de Química Aplicada, Universidad Papaloapan, Campus Tuxtepec, Tuxtepec 68301, Mexico; oramirez@unpa.edu.mx (O.A.R.-M.); kcastro@unpa.edu.mx (K.V.C.-C.); 5Departamento de Genética y Microbiología, Universidad de Murcia, 30100 Murcia, Spain; vgarre@um.es; 6Institute of Hygiene and Medical Microbiology, Medical University Innsbruck, 6020 Innsbruck, Austria; ulrike.binder@i-med.ac.at

**Keywords:** Arf proteins, mitophagy, mucormycosis, oxidative stress, reactive oxygen species

## Abstract

This study analyzed the role of blood serum in enhancing the mitochondrial metabolism and virulence of Mucorales through rhizoferrin secretion. We observed that the spores of clinically relevant Mucorales produced in the presence of serum exhibited higher virulence in a heterologous infection model of *Galleria mellonella*. Cell-free supernatants of the culture broth obtained from spores produced in serum showed increased toxicity against *Caenorhabditis elegans*, which was linked with the enhanced secretion of rhizoferrin. Spores from Mucoralean species produced or germinated in serum showed increased respiration rates and reactive oxygen species levels. The addition of non-lethal concentrations of potassium cyanide and N-acetylcysteine during the aerobic or anaerobic growth of Mucorales decreased the toxicity of the cell-free supernatants of the culture broth, suggesting that mitochondrial metabolism is important for serum-induced virulence. In support of this hypothesis, a mutant strain of *Mucor lusitanicus* that lacks fermentation and solely relies on oxidative metabolism exhibited virulence levels comparable to those of the wild-type strain under serum-induced conditions. Contrary to the lower virulence observed, even in the serum, the ADP-ribosylation factor-like 2 deletion strain exhibited decreased mitochondrial activity. Moreover, spores produced in the serum of *M. lusitanicus* and *Rhizopus arrhizus* that grew in the presence of a mitophagy inducer showed low virulence. These results suggest that serum-induced mitochondrial activity increases rhizoferrin levels, making Mucorales more virulent.

## 1. Introduction

Mucormycosis is an opportunistic fungal infection caused by members of the order Mucorales, with *Rhizopus arrhizus*, *Lichtheimia corymbifera*, and *Mucor circinelloides* being the main causative agents [1,2]. *Mucor lusitanicus*, formerly known as *M. circinelloides f. lusitanicus* [3], is a dimorphic fungus used to study mucormycosis because of the availability of well-established tools for its genetic manipulation [4].

The incidence of mucormycosis is increasing owing to mounting patient populations harboring risk factors such as uncontrolled diabetes mellitus with ketoacidosis, organ transplantation, and hematological cancer [5]. Recently, mucormycosis has been associated with severe cases of COVID-19 worldwide, especially in India [6], where a cumulative mortality rate of 53% was observed 21 days after the admission of patients with COVID-19-associated rhino-orbito-cerebral mucormycosis [7]. Furthermore, mucormycosis has been associated with post-COVID-19 recovery, possibly because of two main factors: the overuse of steroids to treat COVID-19 infection and uncontrolled diabetes as an underlying disease [8].

*M. lusitanicus* grows as yeast or hyphal cells, depending on the culture conditions. At 28 °C, the presence of hexose, an organic nitrogen source, and an anaerobic environment induces yeast development, whereas oxygen promotes mycelial growth regardless of the carbon or nitrogen source [9,10]. Although the hyphal form is considered virulent owing to its increased tissue invasion potential [11,12], detailed knowledge of the infection process remains elusive. For example, *M. lusitanicus* is a Crabtree-positive microorganism that produces fermentative mycelia under high-glucose (≥2%) aerobic culture conditions [10]; the cell-free supernatant of the culture broth (SS) from mycelia grown at these conditions is nontoxic against *Caenorhabditis elegans* [13]. In contrast, the SS from mycelia grown in low-glucose or non-fermentable carbon sources is more toxic [12,13]. Understanding the factors that influence metabolism associated with mycelial development in *M. lusitanicus* is crucial for understanding the virulence process.

Some signaling pathways have been reported to be important for regulating mycelial growth in *M. lusitanicus* and influencing mucormycosis [1]. The calcineurin pathway plays a crucial role in the yeast–hyphal transition. A mutation in *cnbR*, which encodes the regulatory subunit of calcineurin, leads to a yeast-locked morphology even in the presence of oxygen, resulting in avirulence [11]. In contrast, a mutation in *adh1*, which encodes alcohol dehydrogenase 1, generates a monomorphic hyphal strain that cannot perform fermentative metabolism and is more virulent than the wild-type (WT) strain [12]. Hyphal development in *M. lusitanicus* is generally accompanied by an increase in oxidative metabolism [13,14]. Rhizoferrin, produced by rhizoferrin synthetase (Rfs), is positively correlated with virulence, and its synthesis in *M. lusitanicus* is stimulated by enhanced oxidative metabolism [13]. The deletion of *arl2*, which encodes ADP-ribosylation factor-like 2, negatively affects mitochondrial content, decreases oxidative metabolism, and decreases mycelial growth in the presence of a non-fermentable carbon source [15].

We have previously reported that spores generated in the presence of blood serum are more virulent, less susceptible to macrophage-induced death, more resistant to oxidative stress, and exhibit faster hyphal germination under aerobic conditions through an unknown mechanism [16]. Mucormycosis is characterized by angioinvasion and vessel thrombosis [17], which allow the fungus to come into direct contact with serum. Mucormycosis can also occur in tissues with low oxygen levels [18]. This work describes, at the cellular and biochemical levels, the effect of blood serum on virulence through the regulation of mitochondrial homeostasis in Mucorales. This information is relevant because the data from this study suggest that serum can promote the hyphal development of dimorphic species of *Mucor* even at low oxygen levels, which could contribute to the progression of infection during mucormycosis.

## 2. Materials and Methods

### 2.1. Strains and Culture Conditions

Information on the strains used in this study is provided in Appendix A [12,13,15,19,20]. Mucorales were cultured in yeast peptone-glucose (YPG) and yeast-nitrogen-based (YNB) media, as described previously [15]. SSs were obtained after aerobic growth for 48 h at 28 °C via filtration through 0.22 μm filters. Glycerol or oleic acid was used to supplement YNB when a non-fermentable carbon source of 111 mM (equivalent to 2% glucose) was required. When necessary, 0.5 mM of KCN (Sigma, St. Louis, MO, USA), 10 mM of N-acetylcysteine (Sigma), 100 μM of Mdivi-1 (Sigma), and 100 μM of AICAR (Sigma) were added.

Spore production on YPG agar plates, with or without human blood or denatured serum, was performed as described previously [16]. Blood serum was added to the culture medium after sterilization, and the volume was adjusted to preserve its composition in the culture medium. As blood contains approximately 50% serum [21], spores were produced on YPG plates supplemented with 0 (YPG alone), 10, 20, or 40% serum (YPG-10S, YPG-20S, or YPG-40S, respectively) or YPG supplemented with heat-denatured serum at 40% (YPG-40DS).

Liquid cultures were performed at 28 °C on YPG supplemented with 2 or 6% (YPG-2% or YPG-6%, respectively) or on YNB supplemented with 0.1 or 2% (YNB-0.1% or YNB-2%, respectively) glucose. Liquid YPG-2% was supplemented with 10, 20, or 40% of blood serum (YPG-ADD10S, YPG-ADD20S, or YPG-ADD40S, respectively).

### 2.2. Spore Germination

Clinically relevant Mucorales were grown under conditions similar to those previously described for *M. lusitanicus*. Aerobic germination for hyphal growth was performed as previously described [16]. Oxygen levels were determined using an oximeter, as previously described [15]. Low oxygen levels (LOLs) were achieved through constant shaking at 50 rpm at 28 °C in an orbital incubator, which achieved 7.3 ± 0.56% oxygen, or at 40 rpm, which achieved 4.9 ± 0.45% oxygen. *M. lusitanicus* produces a mixture of yeast and hyphal cells at low oxygen levels, as previously reported [15]. Shaking at 150 rpm increased the dissolved oxygen to 14.4 ± 0.64%. Anaerobic growth was performed as previously described [22] to generate yeast cells from *M. lusitanicus*.

After growth in liquid medium, an Olympus CKX41 microscope with a 40× objective lens (Shinjuku, Tokyo, Japan) was used to evaluate fungal germination or used to capture images. Hyphae, yeasts, and swollen spores observed in 100 cells after growth were used to determine the spore germination rate.

### 2.3. Total RNA and DNA Isolation from M. lusitanicus

Total RNA and DNA (mitochondrial and nuclear) from *M. lusitanicus* were isolated using the RNAeasy Mini Kit and QIAamp DNA Mini Kit, respectively (Qiagen, Venlo, The Netherlands), as previously described [23].

### 2.4. Oligonucleotide Design and Quantitative Reverse Transcription Polymerase Chain Reaction (RT-qPCR)

Primers and hydrolysis probes for genes encoding the membrane proton channel domain (F0) subunit (*atp9*), alcohol dehydrogenase 1 (*adh1*), the regulatory subunit of protein kinase A (*pkaR1*), and a subunit of the transcription factor TFIIIC from RNA polymerase III (*tfc-1*, used as a reference gene) were used as previously described [15,23].

### 2.5. Determination of Mitochondrial DNA Abundance Using qPCR

As previously reported, the ratio of mitochondrial (*atp9*) to nuclear (*tfc-1*) DNA served as a measure of mitochondrial abundance in each *M. lusitanicus* strain [15].

### 2.6. Determination of Glucose and Ethanol Levels

The supernatant of a 12 h or 48 h aerobic culture of each strain grown on YPG with 2% glucose was filtered and analyzed using an Aminex HPC-87Ca column (Bio-Rad, Hercules, CA, USA). Glucose consumption and ethanol production were determined using liquid chromatography, as described previously [14]. The operating conditions were as follows: column temperature, 80 °C; deionized water as mobile phase at 0.7 mL/min through 20 min running; injection volume, 25 μL. Quantitation was performed using a refraction index detector based on calibration plots using glucose and ethanol as standard compounds (Sigma-Aldrich, Burlington, MA, USA), obtaining a linear coefficient of determination (R2) of 0.99 for each.

### 2.7. Caenorhabditis Elegans and Galleria Mellonella Killing Assays

*C. elegans* Bristol N2 was grown to obtain worms in the young adult phase, as previously described [24]. The cell-free supernatants of the culture broth (SS) from mycelial growth with the indicated or sterile culture media were co-incubated with the worms for 48–60 h at 18 °C, as described previously [13]. Virulence assays in *G. mellonella* were performed by injecting 20 μL of phosphate-buffered saline containing 5000 spores, which is double the inoculum size based on the LD50 of the virulent strain *R. arrhizus* without serum stimulation (Appendix A) into larvae. Twenty larvae per strain were used in three independent experiments. Survival was monitored every 24 h over eight days.

### 2.8. Relative Quantitation of Rhizoferrin

Rhizoferrin was analyzed using UPLC-ESI-TOF-MS with an ACQUITY UPLC I-Class system (Waters, Republic of Singapore, Singapore) coupled with a Synapt G2-Si mass spectrometer (Waters, London, UK) equipped with an electrospray ionization source, controlled by Waters MassLynx 4.1 software (Waters, London, UK), as described previously [13].

### 2.9. Respiration Measurements

Cell respiration was determined after 6 h of growth at 28 °C in liquid culture media using 5 × 10^5^ spores/mL of the different strains, as described previously [15].

### 2.10. Mitochondrial Membrane Potential and Hydroxyl Radical Quantification

Spores were germinated for 3 h in aerobic conditions or 6 h under anaerobic conditions on YPG liquid culture medium containing 100 nM MitoTrackerTM Green FM (Invitrogen, Walthman, MA, USA) for fluorescent staining of mitochondria to determine mitochondrial membrane potential. The samples were rinsed with water and mounted on microscope slides with 50% (*v*/*v*) glycerol. The samples were observed at a wavelength specific to MitoTrackerTM Green FM using a confocal microscope (FV1000; Olympus, Tokyo, Japan) with a 500–523 nm emission filter (488 nm excitation line). Green fluorescence was quantified as previously described [13,15]. For reactive oxygen species (ROS) (OH^−^), we used 3′-(p-aminophenyl) fluorescein (Thermo Fisher Scientific, CDMX, Mexico) and treated the cells as previously reported [13].

### 2.11. Statistical Analysis

All data were evaluated using analysis of variance (ANOVA; significant differences (*p* < 0.05) are indicated by asterisks; * *p* < 0.05, ** *p* < 0.01, *** *p* < 0.001) using Fisher’s exact test. The Mantel–Cox test was used for survival analysis (*** *p* < 0.01). This trace (⌐) indicates that the statistical analysys was done versus control or WT condition. When the results were not considered significant, an additional indication was not provided (*p* > 0.05).

## 3. Results

### 3.1. Blood Serum Increases the Mitochondrial Activity, ROS Generation, and Virulence of M. lusitanicus

To determine the effect of blood serum concentration on the biological processes evaluated, 10%, 20%, and 40% *v*/*v* concentrations of blood serum (YPG-10S to YPG-40S) were used for the spore production of *M. lusitanicus*. The blood serum concentration used for spore generation correlated with the increased germination rate during aerobic growth in a medium containing the non-fermentable carbon source, oleic acid, as the sole carbon source (Figure 1A), compared with those of the spores produced on denatured serum (YPG-40DS) and YPG alone (Figure 1A). In addition, after growth in glycerol and oleic acid, mycelia from spores produced in serum showed higher *atp9* mitochondrial DNA and transcript levels than mycelia from spores produced on YPG-40DS and YPG (Appendix A). These data suggested that an increase in mitochondrial content could increase mitochondrial activity to allow faster germination in the presence of a non-fermentable carbon source.

To determine a possible connection between serum and increased mitochondrial oxidative metabolism, we analyzed the respiration levels of the mutant strain M5, which is unable to perform fermentation because of a mutation in the *adh1* gene encoding an alcohol dehydrogenase [12], in comparison with the serum-produced spores from the WT strain R7B. After growing aerobically for 3 h on YPG, M5 (*adh1*^−^) spores produced on YPG alone showed 19.9–29.1% lower levels of respiration than germinated spores from the WT R7B strain produced in blood serum (YPG-20S) or spores from the R7B strain growing in the presence of blood serum (YPG-ADD20S) (Figure 1B). In contrast, low respiration levels were observed in germinated spores from R7B produced in YPG alone or denatured blood serum (YPG-20DS) (Figure 1B). These results showed that blood serum increased respiration in the WT strain of *M. lusitanicus* to a greater extent in the M5 strain, which grows mainly through oxidative metabolism, suggesting that blood serum increases mitochondrial metabolism in the WT strain. Moreover, spores produced on YPG-20S and those grown on aerobically germinated YPG-ADD20S had greater mitochondrial membrane potential and hydroxyl radical levels during germination than spores generated on YPG or YPG-20DS (Figure 1C–E), supporting the idea that blood serum increased mitochondrial activity in *M. lusitanicus*.

We have previously described the effect of culture media on the fermentative oxidative metabolism of *M. lusitanicus*, resulting in differences in SS toxicity. For example, a decrease in the toxicity of SS was observed when high (6%) levels of glucose, which promote fermentation, were present in the culture medium instead of the regular amount (2%), whereas the opposite was observed when low glucose (0.1%) concentrations, which induce oxidative metabolism, were used [13]. The SS from M5 showed increased toxicity when obtained from YNB with 0.1% glucose (YNB-0.1%), causing the death of all nematodes after 12 h of interaction (Figure 1F). The toxicity of SS from M5 was partially reduced by high glucose levels when YPG was used at 6% glucose (YPG-6%), comparable to the levels observed with SS produced with YNB-0.1% from the R7B strain, even when the spores were serum-produced (Figure 1F). The SS obtained from R7B grown on YPG-6% lost its toxicity (Figure 1F). The incubation of nematodes with YPG or YNB supplemented with serum prevented the direct observation of nematodes under a stereoscope (Appendix A); for this reason, the observation of nematodes incubated with spores germinating in the presence of serum was not feasible under our conditions. To determine whether the spores, and not only the SS, were involved in virulence, the spores from the R7B strain produced on YPG-20S were inoculated into *G. mellonella*, showing an increased virulence compared with the spores produced on YPG (Figure 1G). The inoculation of mutant M5 spores produced on YPG-20S killed 80% of the larval population at one day post inoculation, whereas the M5 spores produced on YPG alone required two days to ensure similar levels of mortality. In contrast, the spores from R7B produced on YPG-20S required five days to kill the entire population, and the spores produced on YPG at the end of the experiment (eight days) killed 80 ± 7.07% of the population (Figure 1G). Glucose consumption levels after 12 h of aerobic germination of R7B spores produced on YPG-20S were higher than those produced on YPG; however, the germinating spores from M5 produced on YPG-20S showed similar levels of glucose consumption to the germination of R7B spores produced on YPG-20S (Figure 1H). In addition, ethanol production after 12 h was detected only in the SS from the growth of R7B spores produced on YPG, and blood serum decreased its production even after 48 h (Figure 1I). These results show that blood serum increased aerobic hyphal germination in the presence of non-fermentable carbon sources, suggesting that, under aerobic growth conditions, blood serum increases mitochondrial metabolism, suppresses fermentative metabolism, and increases the virulence of *M. lusitanicus*.

### 3.2. M. lusitanicus Spores Cultured in Blood Serum Demonstrate Improved Hyphal Development under Low Oxygen Levels

Hyphal development occurs in low-oxygen environments and is critical during infection. The corresponding experiments on the influence of blood serum on the hyphal growth of *M. lusitanicus* and other clinically important *Mucor* dimorphic species at low oxygen levels (LOLs) were performed to determine if the spores produced in serum showed higher hyphal growth under low oxygen levels (40 rpm, 4.9 ± 0.45% O_2_). Under these conditions, 80 ± 4.5% of the spores from the R7B strain of *M. lusitanicus* that were produced on YPG alone germinated as yeast, while the hyphal cells subsequently decreased (20 ± 4.5%) (Figure 2A). The germination of spores that were produced in serum increased the hyphal cells under low oxygen levels, which was more evident in spores produced on YPG-40S that generated 57.5 ± 3.3% hyphae and 42.5 ± 3.3% yeast (Figure 2A and Appendix A). This effect of serum on germination was not species-specific, as hyphal development under low-oxygen conditions increased when *M. circinelloides* (MC39) and *Mucor racemosus* (MR11) spores were produced on YPG-20S, with increases similar to those observed in *M. lusitanicus* (Appendix A). These increases were directly correlated (R2 = 0.8874) with mitochondrial content in spores germinated under low oxygen levels, particularly in spores produced on YPG-20S and YPG-40S (Figure 2B). Moreover, these cells showed increased transcript levels of the molecular marker of hyphal growth, *pkaR1*, and decreased levels of the fermentative molecular marker, *adh1* [23], depending on the serum concentration used to produce spores (Figure 2C,D).

Spores germinated in the presence of blood serum showed results similar to those produced on a medium supplemented with blood serum. Under low oxygen levels, the presence of blood serum at 20% (YPG-ADD20S) or 40% (YPG-ADD40S) in the culture medium increased hyphal germination up to 82.5 ± 3.44% or 85 ± 8.07%, respectively, whereas spores grown on YPG or in denatured blood serum at 40% (YPG-ADD40DS) generated 22.5 ± 3.1% or 47.5 ± 2.94% of hyphal germination, respectively (Figure 2E). An increase in hyphal germination was associated with an increase in mtDNA content (Figure 2F).

These results show that blood serum enhanced hyphal development under low oxygen levels, which correlated with increased mitochondrial content.

### 3.3. Blood Serum Increases the Mitochondrial Metabolism and Virulence of Clinically Relevant Mucorales

To determine whether blood serum increased the toxicity and virulence of other clinically relevant Mucorales, as *M. lusitanicus* does, the SSs obtained from the aerobic growth of spores produced on YPG-20S from clinically relevant Mucorales were evaluated. The data showed that the SSs from the growth of spores produced on blood serum of *R. arrhizus* (RA12), *M. circinelloides* (MC39), and *L. corymbifera* (LC42) were more toxic to nematodes than those obtained from spores produced on YPG alone (Figure 3A,B). We tested the effects of SSs from Mucorales cultures grown aerobically on YPG-2% and YPG-6% on the spores produced on YPG-20S. We found that growth under high glucose levels (YPG-6%) decreased the toxicity of all SSs from spores produced in the serum, except for the SS from *R. arrhizus*, which only decreased partially (Appendix A).

Respiration and ROS production were measured to determine whether blood serum increased the mitochondrial activity in clinically relevant Mucorales species, as observed in *M. lusitanicus*. The oxygen consumption and ROS levels of aerobically germinating spores produced on YPG-20S from all Mucorales were higher than those of spores obtained on YPG, except for *L. corymbifera*, for which germinating spores produced on YPG-20S exhibited lower oxygen consumption and ROS levels than those obtained on YPG (Figure 3C,D). The addition of the ROS scavenger N-acetylcysteine (10 mM N-Ace) or the mitochondrial electron transfer chain blocker potassium cyanide (0.5 mM KCN) during the aerobic growth of spores produced on YPG-20S from all the Mucorales eliminated the toxicity of the SS obtained (Figure 3E,F) without affecting biomass production (Appendix A). Remarkably, except for *M. lusitanicus*, SSs derived from the anaerobic growth of spores produced in the blood serum of *M. circinelloides*, *R. arrhizus*, and *L. corymbifera* showed higher toxicity than the SSs obtained from spores of Mucorales produced under similar growth conditions in the absence of serum (Figure 3G,H). To determine whether mitochondrial metabolism could be implicated in the increase in serum-induced toxicity of SSs obtained from anaerobic growth, N-ace and KCN were added to the cultures during growth. The results showed a decrease in the toxicity of the SSs obtained from the anaerobic growth of spores produced on YPG-20S in the presence of N-ace and KCN (Appendix A). These results suggest that blood serum stimulates virulence through mitochondrial activity during the anaerobic growth of all clinically relevant Mucorales tested. The increased toxicity of the SS induced by blood serum under anaerobic conditions was not linked to the dimorphism shown by some Mucorales species. *L. corymbifera* produced yeast with a low level of hyphal germination (less than 10%) under anaerobic growth conditions, whereas *R. arrhizus* only showed hyphal development (Appendix A). The mitochondrial activity measured using the MitoTrackerTM Green FM probe after 6 h under anaerobic growth conditions revealed higher activity in the germinating spores produced on YPG-20S than in those produced on YPG alone, except for *M. lusitanicus* (Figure 3I,J).

To determine whether serum affects the spore virulence of clinically relevant Mucorales, the serum-produced spores were injected into *G. mellonella* larvae, which revealed that the spores produced on YPG-20S resulted in a higher mortality than those produced on YPG (Figure 4A,B). Moreover, the spores that were produced on YPG-20S showed faster growth at 36 °C than those that were generated on YPG (Figure 4C). These results indicate that in Mucoralean spores obtained from a medium containing serum, hyphal growth, thermotolerance, SS toxicity (even during anaerobic growth), and virulence increase in correlation with adequate mitochondrial oxidative metabolism and ROS production, indicating the pivotal role of ROS in these processes.

### 3.4. Blood Serum Increases the Rhizoferrin Levels in M. lusitanicus and other Clinically Relevant Mucorales

Rhizoferrin increases in *M. lusitanicus* under growth conditions that promote mitochondrial oxidative metabolism, whereas the deletion of *rfs* leads to an avirulent strain [13]. Therefore, we hypothesized that the virulence induced by blood serum could be due to an increase in rhizoferrin secretion. *M. lusitanicus* MU636 (WT) spores generated on YPG-20S accumulated higher levels of both the *rfs* transcript (Figure 5A) and rhizoferrin (Figure 5B) than spores produced on YPG after aerobic growth. Similarly, SSs obtained from the aerobic growth of spores produced on YPG-20S from *R. arrhizus*, *L. corymbifera*, and *M. circinelloides* accumulated more rhizoferrin than SSs from Mucorales spores produced on YPG alone (Figure 5C). Previous studies have shown that the toxicity of SS against nematodes is decreased through Fe^2+^ supplementation during the aerobic culture of *M. lusitanicus*, which is partially explained by the downregulation of *rfs* [13]. Our results demonstrated that adding iron to the aerobic growth of Mucorales spores produced on YPG-20S lowered the toxicity of SSs to worms (Figure 5D). In addition, ∆*rfs* spores produced on YPG-20S were not virulent to *G. mellonella* (Figure 5E), and SSs produced through aerobic development did not exhibit any toxicity toward the worm (Figure 5F). These findings suggest that blood serum stimulates Mucorales virulence by increasing rhizoferrin production.

### 3.5. Blood Serum Requires Increased Mitochondrial Content for a Virulent Phenotype and Hyphal Growth under Low Oxygen Levels

The small GTPase Arl2 positively regulates mitochondrial content in *M. lusitanicus* [15]. Δ*arl2* showed similar yeast/hyphal proportions as WT under low oxygen levels on YPG alone (Figure 6A). Although hyphal growth under low oxygen levels was stimulated in ∆*arl2* in the spores produced on YPG-20S, this stimulation was lower than that in the WT, indicating that the blood serum did not completely restore hyphal development in ∆*arl2* (Figure 6A).

During aerobic development with a non-fermentable carbon source, ∆*arl2* exhibits reduced hyphal germination compared with the WT strain [15]. Additional studies are needed to investigate the correlation between reduced hyphal growth and mitochondrial activity in germinating spores from ∆*arl2* in blood serum. Oxygen consumption and hydroxyl radical (ROS) levels by the Δ*arl2* germinating spores under aerobic conditions were lower than those of the WT, including the Δ*arl2* germinating spores produced on YPG 20S (Figure 6B,C). The toxicity of SS obtained from the aerobic growth of Δ*arl2* spores produced on YPG-20S was lower than that obtained from the WT (Figure 6D).

To determine whether mitochondrial content influences the virulent phenotype induced in the biological model of *M. lusitanicus* and the most clinically important Mucoral *R. arrhizus*, we used pharmacological modulators of mitophagy named AICAR and Mdivi-1, which induce and suppress mitophagy, respectively, in mammalian cells [25,26]. When spores from *M. lusitanicus* and *R. arrhizus* were produced on YPG-20S and subsequently aerobically grown for 48 h in the presence of AICAR to obtain the corresponding SSs, their toxicity toward *C. elegans* was reduced. However, the toxicity of the SSs increased when Mdivi-1 was present during growth (Figure 6E). In *M. lusitanicus* and *R. arrhizus,* AICAR or Mdivi-1 did not affect biomass after 48 h of aerobic growth compared with the control. Regardless of AICAR or Mdivi-1, blood serum increased biomass production after culture (Appendix A).

Blood serum enhanced the formation of the hyphae during the growth of WT strains of *Mucor* dimorphic species at low oxygen levels (Figure 2 and Appendix A); this was also found when spores from the MU636 WT strain of *M. lusitanicus* produced without serum germinated in the presence of Mdivi-1 under the same growth conditions (Figure 6F and Appendix A). AICAR did not alter the hyphal yeast proportion during the germination of spores produced in the absence of serum compared with the control but decreased the hyphal growth of spores generated in blood serum (Figure 6G and Appendix A). In addition, AICAR led to the shortening of the hyphal length of spores, independent of serum supplementation (Figure 6G and Appendix A). Moreover, Mdivi-1 did not restore the hyphal development of ∆*arl2* spores produced on YPG-20S (Figure 6G,H and Appendix A). In agreement with this, the addition of AICAR or Mdivi-1 during the low-oxygen germination (6 h) of *M. lusitanicus* from spores produced or not in serum led to decreased and enhanced levels of the mitochondrial *atp9* gene, respectively, in comparison to the control (Appendix A). Furthermore, during aerobic growth, the addition of Mdivi-1 or AICAR increased or decreased, respectively, the respiration of all Mucorales (Figure 6H).

These findings support the hypothesis that blood serum promotes virulence in *M. lusitanicus* by increasing mitochondrial oxidative metabolism via Arl2. In addition, the results suggest that mitophagy repression exhibits similar SS toxicity levels in *M. lusitanicus* and *R. arrhizus* as those induced by blood serum.

### 3.6. Effect of N-acetylglucosamine on the Toxicity of Mucorales-Derived SS

Since N-acetylglucosamine enhances *Candida albicans* hyphal growth [27] and virulence [28], Mucorales were examined for a similar effect in its presence. The toxic effects of the SSs obtained after the growth of Mucorales on YPG supplemented with 5 mM and 50 mM N-acetylglucosamine against nematodes were not enhanced compared with the control (Appendix A), in contrast to the enhanced toxicity of SS acquired after the aerobic germination of spores produced on YPG-20S (Appendix A). In contrast to what was observed in *C. albicans*, where hyphal growth was stimulated (Appendix A), N-acetylglucosamine did not promote hyphal growth in *M. lusitanicus* (Appendix A). These findings suggest that molecules other than N-acetylglucosamine drive Mucorales virulence.

## 4. Discussion

Blood serum increases *M. lusitanicus* virulence through an unknown biochemical mechanism [16], whereas mitochondrial oxidative metabolism enhances the virulence of *M. lusitanicus* [13] and increases hyphal development in some dimorphic *Mucor* species [9]. Depending on the culture growth conditions, the hyphal development has been shown to reveal the virulent phenotype in *M. lusitanicus* [13]. The results of the present study support the concept that blood serum enhances hyphal growth, stimulates mitochondrial metabolism by increasing the mitochondrial content, represses fermentative metabolism, and increases rhizoferrin secretion to promote virulence and hyphal development in *M. lusitanicus* and other clinically relevant Mucorales species.

Our findings demonstrate that blood serum increases mitochondrial content and respiration in *M. lusitanicus* and other Mucorales species. Fungal mitochondria control metabolic adaptability and stress responses, contributing to successful development, infection, and disease [29]. Our results suggest that the absence of toxicity of Mucorales SSs obtained after aerobic growth in the presence of KCN or N-ace negatively affected mitochondrial activity and ROS production. These observations support the hypothesis that the Mucorales species studied require adequate mitochondrial activity during aerobic development for virulence.

Blood serum is an oxidative stressor in *M. lusitanicus* and other Mucorales species, such as *R. arrhizus*. Based on previous findings, adding 0.5 mM H_2_O_2_ to *M. lusitanicus* increases its virulence [13]. Blood serum enhanced ROS generation in the Mucorales examined (except *L. corymbifera*), which increased SS toxicity and spore virulence. Conversely, N-ace reduced the toxicity of SS in all Mucorales species, suggesting that the blood serum produces ROS that increase virulence through unknown mechanisms.

Fungal pathogens produce several ROS such as singlet oxygen (^1^O^–2^), superoxide anions (O^–2^), and hydroxyl radicals (OH^–^) [30]. Mitochondria and NADPH oxidase (NOX) are involved in ROS production [31]. Mitochondrial ROS may increase NOX ROS production and vice versa [31]. In *C. albicans*, NOX forms a superoxide gradient outside the cell and is converted to H_2_O_2_ by superoxide dismutase Sod5, which enhances hyphal development and pathogenicity [32]. Blood serum increases ROS levels in *C. albicans* by regulating cAMP levels [33]. Our results showed that blood serum increased the OH^–^ radical levels in Mucorales (excluding *L. corymbifera*). Further research is required to determine the ROS type and origin that affect the Mucorales spore-to-hyphae switch and how this affects virulence potential.

Mitochondria are the key organelles involved in iron homeostasis [34]. In *M. lusitanicus*, the overexpression of the non-ribosomal siderophore enzyme Rfs, which synthesizes rhizoferrin, increases SS toxicity. However, the addition of N-ace and KCN eliminated SS toxicity in this strain [13]. This may be explained by the mitochondrial origin of the substrates (citrate and diaminobutane) used for rhizoferrin synthesis. Our results showed that spores of ∆*rfs* produced in blood serum were avirulent against *G. mellonella*, and the SS produced during the aerobic germination of these spores was nontoxic to *C. elegans*. In *M. lusitanicus* and other Mucorales species, blood serum increases rhizoferrin production. Rhizoferrin may scavenge iron from the medium, making it available to the fungus and store iron intracellularly [34]. In addition, rhizoferrin may provide iron to iron–sulfur cluster ensembles such as *Erwinia chrysanthemi* siderophores [35]. These activities may increase the mitochondrial oxidative metabolism and pathogenicity in Mucorales.

In contrast, high glucose levels stimulate fermentative metabolism in dimorphic *Mucor* species [9]; in *M. lusitanicus*, this stimulus increases cAMP levels and PKA activation, and eliminated SS’s toxicity reached during aerobic growth [13]. This effect seems to apply to other Mucorales, except *R. arrhizus*; during the aerobic growth of this species, high glucose levels partially decrease SS toxicity from the mycelial growth of spores produced in the blood serum. Moreover, anaerobic culture enhanced the fermentative metabolism in *R. arrhizus* [36], and the SSs produced during the anaerobic growth of Mucorales exhibited no toxicity. In addition, SSs from Mucorales obtained from the anaerobic growth of spores produced in blood serum were toxic to nematodes. However, increasing the glucose levels and the addition of KCN and N-ace repressed this toxicity under similar growth conditions. These results suggest that blood serum may improve some mitochondrial pathways that promote SS toxicity during the anaerobic growth of Mucorales, considering that the Krebs cycle generates organic acids, such as citric acid, in *S. cerevisiae* under anaerobiosis [37]. In addition, blood serum at 12 h of growth inhibited aerobic fermentative metabolism in the WT strains of *M. lusitanicus* to levels similar to those in the *adh1*^−^ mutant strain (M5), which lacks fermentative metabolism and has a virulent phenotype [12]. A deletion strain of *arl2*, which reduces mitochondrial content and enhances fermentative metabolism [15], was used to determine whether blood serum signaling requires mitochondrial activity. Blood serum did not restore the virulence of the mitochondrial-deficient ∆*arl2* strain. Aerobic germinating spores from ∆*arl2* produced in blood serum had a decreased respiration rate, ROS generation, and virulence, indicating that optimum mitochondrial function is needed for a proper response to blood serum. Arf proteins are involved in mitochondrial homeostasis because the lack of Arf1 results in mitophagy defects similar to those observed in *S. cerevisiae* and mammals [38]. At the molecular level, how Arl2 controls mitochondrial homeostasis in *M. lusitanicus* remains unknown. Our results suggest that during aerobic growth, the mitophagy inhibitor Mdivi-1 stimulates the SS toxicity of the WT strains of *M. lusitanicus* and *R. arrhizus*, whereas AICAR represses the toxicity of the SSs from both Mucorales, even if the spores are produced in blood serum (Figure 6E). During the aerobic growth of *M. lusitanicus*, mitochondrial *atp9* gene levels increased and decreased in the presence of Mdivi-1 and AICAR, respectively. The stimulation of toxicity induced by Mdivi-1 was not observed in the ∆*arl2* of *M. lusitanicus*; it has been proposed that Arl2 is a vesicle biogenesis regulator that may be involved in organelle biogenesis [15]. These results strongly suggest that blood serum increases mitochondrial activity, probably through an increase in mitochondrial content. Moreover, in *M. lusitanicus,* blood serum, through Arl2, may control other regulators of mitochondrial dynamics, such as biogenesis, fusion, fission, and/or mitophagy, which need to be explored further in Mucorales.

The absence of fermentation in the *M. lusitanicus* M5 strain, which depends on oxidative metabolism, correlates with higher virulence in the blood serum at levels comparable to those of SS from known virulent Mucorales species, such as *R. arrhizus* and *L. corymbifera* [39]. This finding suggests that Mucorales species associated with mucormycosis may have a greater intrinsic mitochondrial metabolism than species such as *M. lusitanicus*. This hypothesis is supported by the fact that SS generated during the aerobic growth of the *M. lusitanicus* M5 (*adh1-*) strain and WT *R. arrhizus* on YPG containing 6% glucose, which stimulates fermentation, exhibited a lower, albeit not null, SS toxicity, compared to the other Mucorales, which exhibited no SS toxicity under these conditions. It is likely that *R. arrhizus* does not rely on fermentative metabolism during aerobic development, at least not as much as dimorphic *Mucor* species under the same growth conditions. *M. lusitanicus*, *M. racemosus*, and *M. circinelloides* grew a mixture of yeast and hyphal cells, and exhibited serum-stimulated hyphal growth with increased mitochondrial content. Hyphal growth has been observed in histological samples of dimorphic *Mucor* species in patient tissues [40,41]. Our results may explain the prevalent hyphal morphology of dimorphic Mucor species during mucormycosis, which can invade different tissues with low oxygen levels such as the human peritoneal cavity (5.3% oxygen), spleen (3–4% oxygen), and the intestinal tissue (7% oxygen) [42,43] or wounds (approximately 1% oxygen) [18]. During mucormycosis, molecules from blood serum may induce the hyphal development of Mucor in tissues. Blood serum may trigger the angioinvasive spread by regulating *R. arrhizus* gene expression, as described for *M. lusitanicus*, in which blood serum increases *cotH3* mRNA levels. CotH facilitates host cell invasion by interacting with human GRP78, which is highly expressed in diabetes [44]. The ability to grow at temperatures ≥37 °C is a major feature of the pathogenicity of mammalian fungi. Blood serum spores showed increased growth at 36 °C in the Mucorales tested. Deleting the manganese superoxide dismutase (*sod2*) gene in *Cryptococcus neoformans* decreases aerobic but not anaerobic growth at 37 °C, suggesting that mitochondrial activity may affect this process [45]. Similarly, blood serum enhanced Mucorales growth at 36 °C by increasing mitochondrial oxidative metabolism.

In contrast to *C. albicans,* the morphogenic action of N-acetylglucosamine, which is present in blood serum, increases hyphal growth and pathogenicity in *C. albicans* [30]. *Mucor lusitanicus* hyphal development and SS toxicity were unaffected by N-acetylglucosamine. The primary limitations of this study are the unknown blood serum molecules that cause the virulent phenotype of Mucorales and nearly all the signaling pathways regulated by these serum molecules, with the exception of Arl2 and Rfs, which participate in the virulence increase in *M. lusitanicus*.

## 5. Conclusions

In conclusion, under our experimental conditions, blood serum increased Mucorales virulence by enhancing mitochondrial activity, ROS production, and rhizoferrin synthesis, probably through the repression of mitophagy; in the case of *M. lusitanicus*, these processes require Arl2 and Rfs.

## Figures and Tables

**Figure 1 jof-09-01127-f001:**
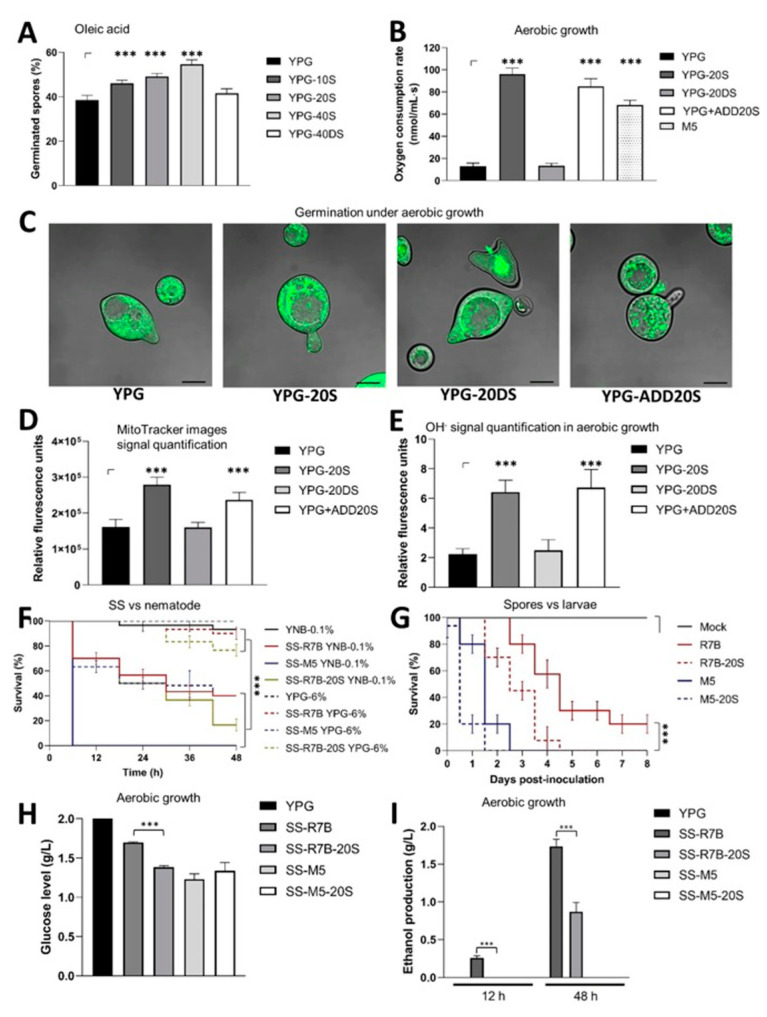
Effect of blood serum on mitochondrial oxidative metabolism and virulence in *M. lusitanicus*. Spores from the WT R7B strain produced on yeast peptone glucose (YPG) or YPG supplemented with 10 (YPG-10S), 20 (YPG-20S), and 40% (YPG-40S) blood serum or 40% denatured blood serum (YPG-40DS) were aerobically germinated for 4 h in liquid-yeast-nitrogen-based (YNB) medium supplemented with 111 mM oleic acid instead of glucose to determine the (**A**) germination rate. Spores produced on YPG, YPG supplemented with 20% blood serum (YPG-20S), or YPG 20% denatured blood serum (YPG-20DS), or spores growing in the presence of 20% blood serum (YPG-ADD20S) were germinated for 3 h on YPG; (**B**) oxygen consumption was quantified; and (**C**) fluorescence was visualized in the presence of MitoTrackerTM Green FM using confocal microscopy (120×); bars = 20 μm. (**D**) The fluorescence was quantified using the MitoTrackerTM Green FM signal and (**E**) hydroxyl radical (OH^-^) concentration. (**F**) SSs obtained after 48 h of growth on YNB supplemented with 0.1% glucose (YNB-0.1% in the continuous line) or YPG supplemented with 6% glucose (YPG-6% in the discontinuous line) from spores that were produced on the YPG of the R7B and M5 (*adh1^−^*) strains and assayed against *C. elegans*. A total of 15–20 nematodes were used per experiment and incubated at 18 °C for 48 h. (**G**) Spores (5000) from the *M. lusitanicus* strains produced with or without blood serum were inoculated in *G. mellonella*; 20 larvae were used per assay. Data were statistically analyzed using the Mantel–Cox test; *** *p* < 0.01. (**H**) Glucose consumption and (**I**) ethanol production were determined after 12 or 48 h of aerobic growth from spores of *M. lusitanicus* strains grown on YPG supplemented with blood serum (YPG-20S) or YPG alone. The results are presented as average values obtained from four independent experiments. Data were statistically analyzed using ANOVA with Fisher’s exact test; *** *p* < 0.001.

**Figure 2 jof-09-01127-f002:**
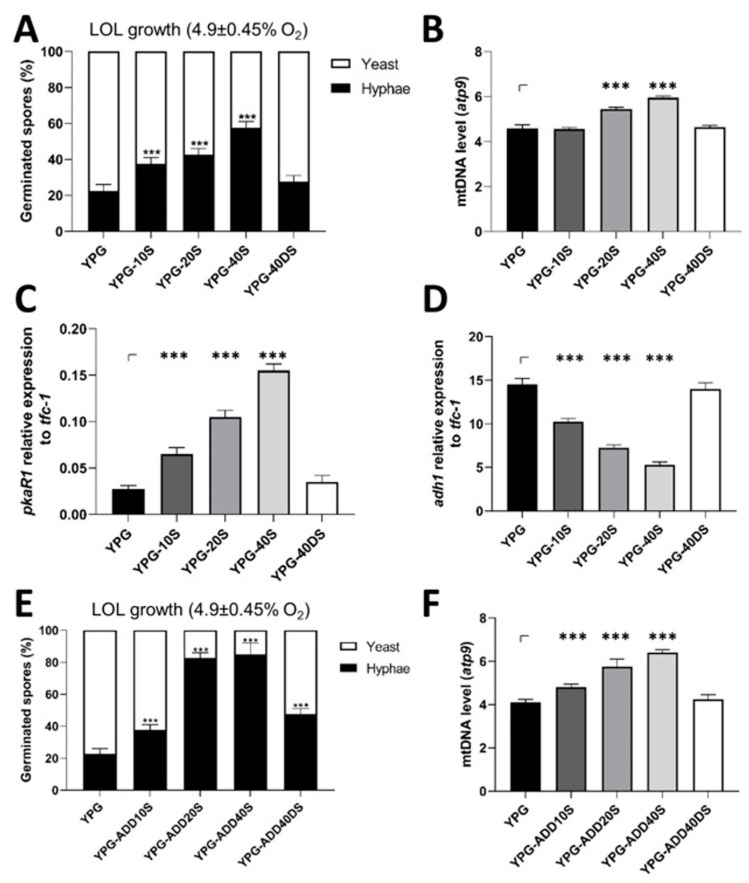
Effect of blood serum on spore germination of *M. lusitanicus* under low oxygen levels (LOLs). Spores from the WT R7B strain of *M. lusitanicus* produced on YPG supplemented with 10, 20, or 40% (YPG-10S to YPG-40S) or without blood serum and denatured blood serum (YPG-40DS) were germinated for 6 h under low oxygen levels (4.9 ± 0.45% oxygen). (**A**) Spores germinating as hyphae and yeast were identified. (**B**) Total DNA was obtained from germinating spores after 6 h (panel (**A**)) and mitochondrial *atp9* levels were quantified using qPCR using *tfc-1* as a genomic DNA marker. Total RNA was used for quantification using RT-qPCR for (**C**) *pkaR1* mRNA (a hyphal growth molecular marker) and (**D**) *adh1* mRNA (a molecular marker for fermentation). A 2-∆Ct analysis was performed to quantify and compare the mRNA levels between samples using *tfc-1* as a reference gene for expression. (**E**) Germination of spores in the presence of different concentrations of blood serum (YPG-ADD10S to YPG-ADD40S) or with denatured blood serum at 40% (YPG-ADD40DS) was determined after 6 h of growth in liquid YPG under low oxygen levels (4.9 ± 0.45% oxygen). (**F**) The expression of mitochondrial *atp9* was quantified using qPCR in cells generated under growth conditions (panel (**E**)). The results are presented as the average of the values obtained from four independent experiments. Data were statistically analyzed using ANOVA and Fisher’s exact tests. *** *p* < 0.001.

**Figure 3 jof-09-01127-f003:**
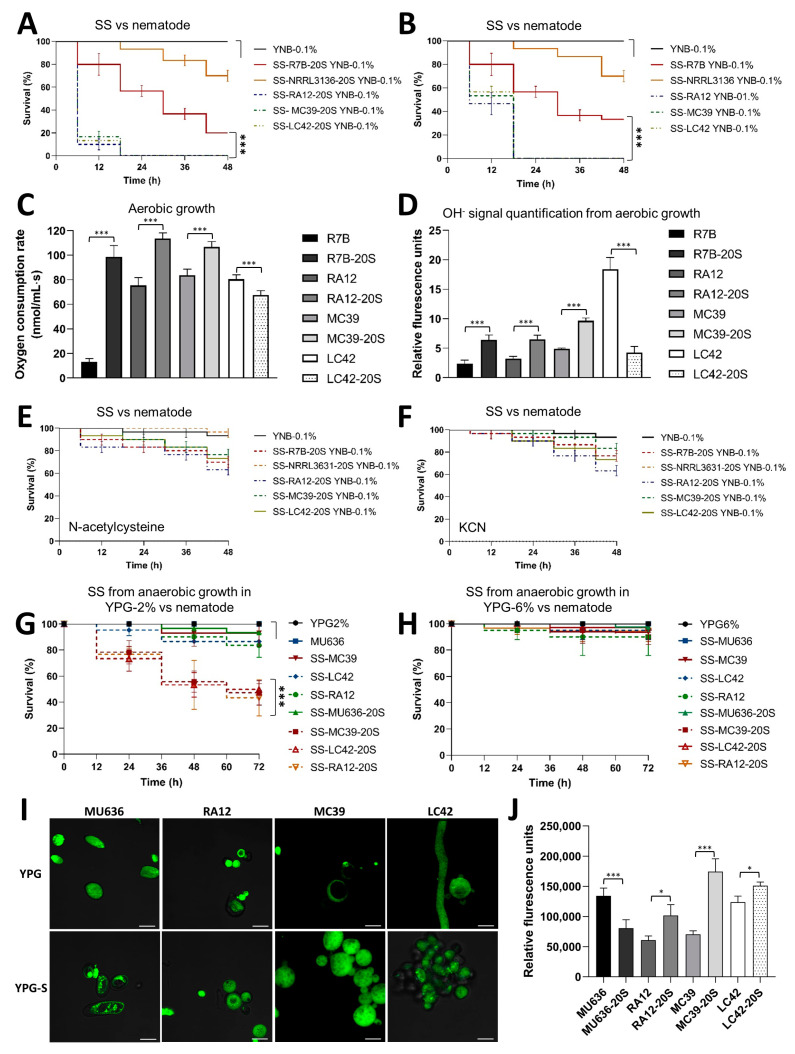
Effect of blood serum on the mitochondrial oxidative metabolism and virulence of Mucorales. SSs were obtained after 48 h of aerobic growth from different Mucorales (**A**) from spores that were produced on YPG supplemented with blood serum at 20% (-20S) and (**B**) from spores that were produced on YPG alone and assayed against *C. elegans*. Spores from the Mucorales produced on YPG supplemented with 20% blood serum (-20S) and produced on YPG alone were aerobically germinating spores after 6 h of growth in liquid YPG to quantify (**C**) oxygen consumption and (**D**) the fluorescence signal of the hydroxyl radical (OH^-^). Toxic effect of the SSs after the addition of (**E**) 10 mM N-acetylcysteine and (**F**) 0.5 mM KCN on YNB-0.1% during the aerobic growth for 48 h of spores that were produced on YPG-20S of the different Mucorales assayed against *C. elegans*. SSs obtained after 24 h of anaerobic growth in (**G**) YPG-2% and (**H**) YPG-6% from the different Mucorales from spores that were produced on YPG and spores that were produced on YPG-20S (-20S) and assayed against *C. elegans*. A total of 15–20 nematodes were used per experiment and incubated at 18 °C for 48 h. (**I**) The fluorescence was visualized in the presence of MitoTrackerTM Green FM using confocal microscopy (120×) from germinating spores after 6 h of anaerobic growth (bars = 20 μm) (**J**) and the fluorescence was quantified using the MitoTrackerTM Green FM signal. The results are presented as average values obtained from four independent experiments. Data were statistically analyzed using ANOVA with Fisher’s exact test; * *p* < 0.05, *** *p* < 0.001. Survival data were statistically analyzed using the Mantel–Cox test; *** *p* < 0.01.

**Figure 4 jof-09-01127-f004:**
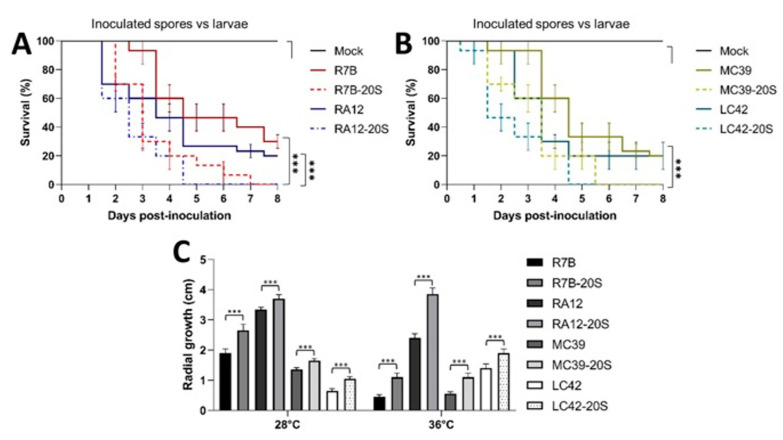
Effect of blood serum on the virulence of Mucorales spores against *G. mellonella*. Spores (5000) from (**A**) *M. lusitanicus* (R7B) and *R. arrhizus* (RA12) or (**B**) *M. circinelloides* (MC39) and *L. corymbifera* (LC42) strains that were produced on YPG supplemented with 20% blood serum (-20S) or on YPG were used to infect *G. mellonella*. Twenty larvae were used in each assay. Three independent experiments were performed. (**C**) Spores of different Mucorales that were produced on YPG supplemented with blood serum (-20S) and on YPG were inoculated (100 spores per plate) to monitor the radial growth at 28 °C and 36 °C. The results are presented as average values obtained from four independent experiments. Survival data were statistically analyzed using the Mantel–Cox test; *** *p* < 0.01. Data were statistically analyzed using ANOVA with Fisher’s exact test; *p* < 0.01; *** *p* < 0.001.

**Figure 5 jof-09-01127-f005:**
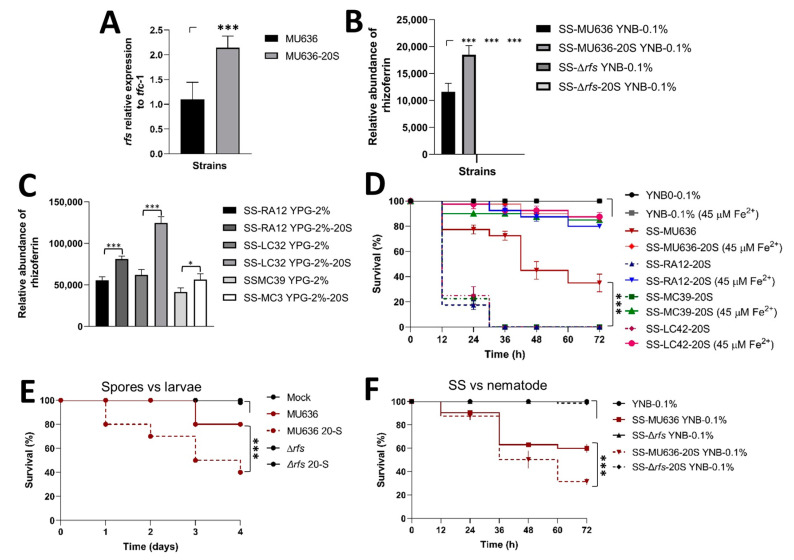
Blood serum upregulates *rfs* and rhizoferrin production in *M. lusitanicus* and clinically relevant Mucorales. (**A**) Total RNA was isolated from MU636 (WT) spores produced on YPG or YPG-20S (-20S) for quantitative analysis of *rfs* transcripts. A 2-∆Ct analysis was performed to quantify and compare the mRNA levels between samples using *tfc-1* as a reference gene for expression. (**B**) Quantification of rhizoferrin in SSs obtained after aerobic growth for 48 h of the corresponding strains of *M. lusitanicus*, MU636 and ∆*rfs,* or (**C**) *R. arrhizus*, RA12; *L. corymbifera*, LC32; and *M. circinelloides*, MC39 on YNB-0.1% from spores obtained on YPG or YPG-20S. (**D**) Toxicity of SSs obtained after aerobic growth for 48 h on YNB medium with 0.1% glucose (YNB-0.1%) supplemented with 45 μM Fe^2+^ from Mucorales spores produced in the blood serum. (**E**) Spores (5000) from the WT and ∆*rfs* strains produced on YPG supplemented with blood serum (-20S) or on YPG were used to infect *G. mellonella*. Twenty larvae were used per assay and three independent experiments were performed. (**F**) SSs obtained after 48 h of aerobic growth from spores of WT and ∆*rfs* of *M. lusitanicus* that were produced on YPG or YPG-20S (-20S) and assayed against *C. elegans*. The results are presented as average values obtained from four independent experiments. Survival data were statistically analyzed using the Mantel–Cox test; *** *p* < 0.01. Data were statistically analyzed using ANOVA with Fisher’s exact test; * *p* < 0.05; *** *p* < 0.001.

**Figure 6 jof-09-01127-f006:**
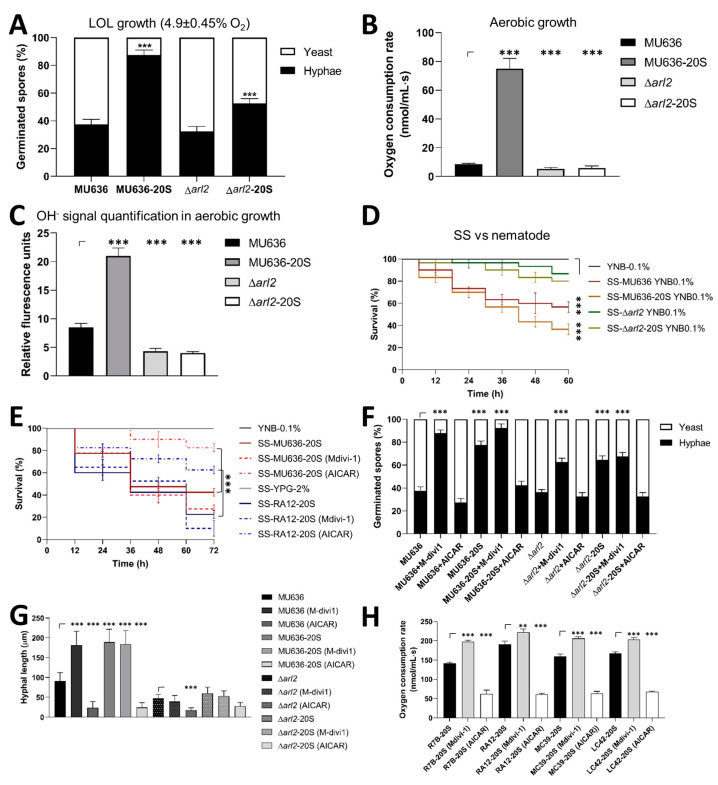
Blood-serum-mediated mitochondrial content in Mucorales. Spores of the MU636 wild-type (WT) and Δ*arl2* strains were produced on YPG alone or supplemented with 20% blood serum (MU636-20S and Δ*arl2*-20S), and afterward, (**A**) the germination of spores was determined after 6 h of growth under low oxygen levels (LOLs) (4.9 ± 0.45% oxygen). Germinating spores after 6 h of aerobic growth (panel (**A**)) were used to quantify (**B**) oxygen consumption and (**C**) the fluorescence signal of hydroxyl radicals (OH^−^). (**D**) SSs were obtained on YNB with 0.1% glucose (YNB-0.1%) from the growth for 48 h of spores produced on YPG supplemented with 20% blood serum (-20S) or from spores produced on YPG alone, or (**E**) 100 μM AICAR or 100 μM Mdivi-1 during growth and assayed against *C. elegans*. A total of 15–20 nematodes were used per experiment and incubated at 18 °C for 60 h. Hyphae and yeast-germinating proportion of spores from MU636 and ∆*arl2* strains after growth for 6 h under low oxygen levels (4.9 ± 0.45%) in liquid YPG-2% medium supplemented with (**F**) 100 μM Mdivi-1 and 100 μM AICAR. (**G**) Hyphal length was registered from the hyphae generated in panel **F**. (**H**) Oxygen consumption was quantified from 3 h of aerobically germinating spores produced in blood serum (-20S) in the presence or absence of 100 μM AICAR or 100 μM Mdivi-1. The results are presented as average values obtained from four independent experiments. Survival data were statistically analyzed using the Mantel–Cox test; *** *p* < 0.01. Data were statistically analyzed using ANOVA with Fisher’s exact test; ** *p* < 0.01, *** *p* < 0.001.

## Data Availability

The data generated or analyzed during this study are fully provided within the published article and its Appendix A.

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
