# Peer review of "Blood Serum Stimulates the Virulence Potential of Mucorales through Enhancement in Mitochondrial Oxidative Metabolism and Rhizoferrin Production"

_jof, 2023, doi:10.3390/jof9121127_

Round 1
Reviewer 1 Report
Comments and Suggestions for Authors
It is an attractive, well-written work rich in details and deals with a suitable subject for the Journal of Fungi. I noted that the authors sought a blood serum product that could molecularly explain the effect on virulence when using blood serum to cultivate the fungus—an extensive and well-applied methodology needed to unravel the mystery, but without success. Although I am not an expert, this virulence must be in conjunction with the product secreted by the fungus during its development. It could be something molecular, linked to protein degradation, or another serum product. However, that would change the culture medium, signaling, and reorganize its metabolic machinery, becoming virulent—like adjusting the extracellular pH necessary for its full development. It is what I believe.
Comments on the Quality of English LanguageDespite not being a native English speaker, I recommend minor revisions concerning the text grammar. A visit to Grammarly Premium would be enough to soften the text.
Author Response
Reviewer #1:
- It is an attractive, well-written work rich in details and deals with a suitable subject for the Journal of Fungi.
Answer 1. We thank to the reviewer for this comment.
- I noted that the authors sought a blood serum product that could molecularly explain the effect on virulence when using blood serum to cultivate the fungus—an extensive and well-applied methodology needed to unravel the mystery, but without success.
Answer 2. We appreciate the reviewer for this comment, the main objective of this work describes, at the cellular and biochemical levels, the effect of blood serum on virulence through the regulation of mitochondrial homeostasis in Mucorales (page 2, lines 90-92).
In addition, we tested the effect of the molecule that is found in serum (N acetylglucosamine) that promotes the virulent phenotype in Candida albicans, but Mucorales were not affected by this molecule. We are currently investigating the molecule in the blood serum responsible for the increase in the virulent phenotype of the Mucorales.
- Although I am not an expert, this virulence must be in conjunction with the product secreted by the fungus during its development. It could be something molecular, linked to protein degradation, or another serum product. However, that would change the culture medium, signaling, and reorganize its metabolic machinery, becoming virulent—like adjusting the extracellular pH necessary for its full development. It is what I believe.
Answer 3. We thank the reviewer for this observation, and indeed we are in agreement with this comment, the blood serum induce the virulence of Mucorales correlating with the secretion of rhizoferrin during the oxidative metabolism (Fig. 5). Rhizoferrin has been reported in a previous report by our group (Alejandre-Castañeda, V., Patiño-Medina, J.A., Valle-Maldonado, M.I., Nuñez-Anita, R.E., Santoyo, G., Castro-Cerritos, K.V., Ortiz-Alvarado, R., Corrales-Escobosa, A.R., Ramírez-Díaz, M.I., Gutiérrez-Corona, J.F., López-Torres, A., Garre, V., Meza-Carmen, V. Secretion of the siderophore rhizoferrin is regulated by the cAMP-PKA pathway and is involved in the virulence of Mucor lusitanicus. Sci Rep. 2022, 12(1), 10649) is necessary for virulent phenotype in M. lusitanicus. The still elusive molecule in the blood serum is inducing the oxidative metabolism, in part by inhibiting the mitophagy, increasing the rhizoferrin synthesis which increases the virulence. As long as we know rhizoferrin do not change the pH, only alters the iron availability, which produce an alteration in the fungal oxidative performance.
Reviewer 2 Report
Comments and Suggestions for Authors
The authors present their study on the the potential virulence of Mucorales in a preclinical model which can be easily clinically relevant. This is an interesting topic since Mucorales is a very life threatening infection in humans and the understanding of the mechanism of invasion is a crucial point to combat infection.
Many statements during the paper are based just on hypotheses, so sometimes this is a limitation of the paper. Although the conclusions are important, many conclusions are based on "likely to be" or "it is supposed to be", therefore a limitation pharagraph needs to be added. The paper is quite difficult to read for clinicians, except for the abstract. Probably as I clinician reading the title I expected more clinically relevant information and considerations.
I would suggest to remove lines 528-537, which also reported some results about "data not shown" and no possible conclusions.
Line 541: could be deleted reading the paper
Lines 577-582: the concept expressed is not clear: is the difference between "lowering" and "partially decrease"?
Comments on the Quality of English LanguageThe English is globally well constructed. I would prefer more linear and simple sentences especially in the discussion.
Globally acceptable.
Author Response
Reviewer #2:
The authors present their study on the the potential virulence of Mucorales in a preclinical model which can be easily clinically relevant. This is an interesting topic since Mucorales is a very life threatening infection in humans and the understanding of the mechanism of invasion is a crucial point to combat infection.
- Many statements during the paper are based just on hypotheses, so sometimes this is a limitation of the paper. Although the conclusions are important, many conclusions are based on "likely to be" or "it is supposed to be", therefore a limitation pharagraph needs to be added.
Answer 1. We thank the reviewer for this suggestion, and in agreement we included a statement in page 17, lines 1143-1146.
- The paper is quite difficult to read for clinicians, except for the abstract. Probably as I clinician reading the title I expected more clinically relevant information and considerations.
Answer 2. We thank for this comment. The main objective of this work describes, at the cellular and biochemical levels, the effect of blood serum on virulence through the regulation of mitochondrial homeostasis in Mucorales (page 2, lines 90-92). Our work contributes to basic science rather than clinical research, but the data we have collected may be useful in the future for understanding and managing mucormycosis.
- I would suggest to remove lines 528-537, which also reported some results about "data not shown" and no possible conclusions.
Answer 3. We thank to the reviewer for this suggestion, we modified the text and location of it in the discussion (Page 17, lines 657-660), we consider is relevant to inform that contrary to Candida albicans, Mucorales did not increased the virulence by the N-acetylglucosamine presence, which offer new hypothesis to be tested by other research groups to identify the still unknow molecule responsible of the increase of the virulence in Mucorales.
- Line 541: could be deleted reading the paper.
Answer 4. We delete this line.
- Lines 577-582: the concept expressed is not clear: is the difference between "lowering" and "partially decrease"?
Answer 5. We thank the reviewer for this observation, and we use the same word to avoid confusion.
- THE ENGLISH IS GLOBALLY WELL CONSTRUCTED.I WOULD PREFER MORE LINEAR AND SIMPLE SENTENCES SPECIALLY IN THE DISCUSSION.
Answer 6. We include a new English edition (from https://app.editage.com/?type=individual) to be more concise in the discussion section.
Reviewer 3 Report
Comments and Suggestions for Authors
The work by Patiño-Medina et al. is very well conducted, prepared and written. The Results are supported by a great amount of nice data showing the involvement of mitochondrial function and of the molecule rhizoferrin in the virulence-enhancing effect caused by blood serum. This is a nice contribution to the field of molecular mycology. I have only a few suggestions to improve the manuscript and some questions: 1. When dealing with human pathogens, I believe it is important to describe the temperatures associated with culture conditions. Preferable, culture conditions should match as much as possible those in the host. However, in order to obtain the desired morphological phase in vitro, researchers have used the tools that they have in their hands. Taking that kind of thought into consideration, can you please add temperature information in lines 61-62 and where else you find more necessary? Fungal growth in subcutaneous tissues occurs at 36-37oC or are these areas a little warmer? When you use 2% in the culture media, to what extent are they growing in higher glucose conditions than in host tissues? The conditions you described for low O2 levels are very interesting to follow. Are these percentages found in other areas than the human peritoneal cavity and wounds (as mentioned in Discussion)? 2. Could you repeat the definition of SS in Materials and Methods? 3. Please define LOL in the legend of Fig. 2 and Fig. 6. 4. Line 420: Please use toxicity or damage, for e.g., to refer to the supernatant effect. In addition, although deletion of the rhizoferrin gene resulted in sterile spores in your Galleria model, other virulence genes might possibly be upregulated by blood serum. Maybe it would be safer to say that blood serum stimulates Mucorales virulence specially by increasing rhizoferrin under your experimental conditions.5. Do you plan to use a mammalian experimental model in further experiments?
Author Response
Reviewer #3:
- The work by Patiño-Medina et al. is very well conducted, prepared and written. The Results are supported by a great amount of nice data showing the involvement of mitochondrial function and of the molecule rhizoferrin in the virulence-enhancing effect caused by blood serum. This is a nice contribution to the field of molecular mycology.
Answer 1. We really appreciate the comments from the reviewer.
I have only a few suggestions to improve the manuscript and some questions: When dealing with human pathogens, I believe it is important to describe the temperatures associated with culture conditions. Preferable, culture conditions should match as much as possible those in the host. However, in order to obtain the desired morphological phase in vitro, researchers have used the tools that they have in their hands. Taking that kind of thought into consideration,
- can you please add temperature information in lines 61-62 and where else you find more necessary?
Answer 2. Thanks for the comment. The information was added.
- Fungal growth in subcutaneous tissues occurs at 36-37oC or are these areas a little warmer?
Answer 3. Thanks for the comment. Yes, the subcutaneous fungal infections occur in zones of the body near to 36°C. The cooler zones of the body surface are the face and neck. For example, the neck with 35°C, then the forehead with 34.5°C, followed by the nose with 33.5°C and the cheeks with 33.2°C.
- When you use 2% in the culture media, to what extent are they growing in higher glucose conditions than in host tissues?
Answer 4. Thanks for the comment. As far as we know, nobody has compared the growth rate in vitro versus in vivo. We know that in vitro in aerobic condition in rich media YPG (2% glucose) the WT strain (MU636) produced 100 % of hyphae after 6 h of incubation. Meanwhile in YNB at 0.1% glucose takes longer time than YPG (2%). In tissues we do not know how long does it takes to the spore population reach 100% germination, but we know that for those dimorphic Mucor species, the prevalent morphology is hyphae, independent of the tissue infected.
- The conditions you described for low O2 levels are very interesting to follow. Are these percentages found in other areas than the human peritoneal cavity and wounds (as mentioned in Discussion)?
Answer 5. Thanks for the comment. Yes, there are other areas with similar concentrations of oxygen (3-8 % oxygen). For example, the spleen can range from 3-4 % oxygen and the intestinal tissue is approximately 7 % oxygen. This information was included in the discussion.
- Could you repeat the definition of SS in Materials and Methods?
Answer 6. Thanks for the comment. The information was added in the Materials and Methods section.
- Please define LOL in the legend of Fig. 2 and Fig. 6. 4.
Answer 7. Thanks for the comment. The information was added.
- Line 420: Please use toxicity or damage, for e.g., to refer to the supernatant effect.
Answer 8. Thanks for the comment. This change was done.
- In addition, although deletion of the rhizoferrin gene resulted in sterile spores in your Galleria model, other virulence genes might possibly be upregulated by blood serum.
Answer 9. Thanks for the comment. Indeed, Mucor lusitanicus could express several virulence factors, but the deletion of rfs led to a dominant negative effect on the global virulent phenotype, at least in all the conditions tested so far. These results suggest that the function of the other virulence factor depends in somehow by Rfs/rhizoferrin. Drfs strain lost all of its virulent phenotype when it was assayed in diabetic-mice, and nematodes models (Alejandre-Castañeda, V., Patiño-Medina, J.A., Valle-Maldonado, M.I., Nuñez-Anita, R.E., Santoyo, G., Castro-Cerritos, K.V., Ortiz-Alvarado, R., Corrales-Escobosa, A.R., Ramírez-Díaz, M.I., Gutiérrez-Corona, J.F., López-Torres, A., Garre, V., Meza-Carmen, V. Secretion of the siderophore rhizoferrin is regulated by the cAMP-PKA pathway and is involved in the virulence of Mucor lusitanicus. Sci Rep. 2022, 12(1), 10649).
- Maybe it would be safer to say that blood serum stimulates Mucorales virulence specially by increasing rhizoferrin under your experimental conditions.
Answer 10. We appreciate this comment from the reviewer.
We include this change in the conclusion.
In conclusion, under our experimental conditions blood serum increased Mucorales virulence by enhancing mitochondrial activity, ROS production, and rhizoferrin synthesis, probably through the repression of the mitophagy, and in the case of M. lusitanicus, these processes require Arl2 and Rfs.
- Do you plan to use a mammalian experimental model in further experiments?
Answer 11. Thanks for the comment. Yes, we are planning to use diabetic mice model in further experiments, once we know the molecule from the blood serum that is implied in the virulence increase.
Round 2
Reviewer 2 Report
Comments and Suggestions for Authors
The authors reviewed the manuscript and I find it now at a good balance between compehension and reporting of scientific results. I just did not understand the sentence at line 630, probably it is just a kind of typing error. Moreover at line 618 the sentence goes to the new line before ending.